# Assessment of the Influence of Canopy Morphology on Leaf Area Density and Drag Coefficient by Means of Wind Tunnel Tests

Shahad Hasan Flayyih Al-Rikabi [1], Enrica Santolini [1], Beatrice Pulvirenti [2], Alberto Barbaresi [1], Daniele Torreggiani [1], Patrizia Tassinari [1] and Marco Bovo [1,*]

1   Department of Agricultural and Food Sciences, University of Bologna, Viale G. Fanin 48, 40127 Bologna, Italy; shahadhasan.alrikabi@unibo.it (S.H.F.A.-R.); enrica.santolini2@unibo.it (E.S.); alberto.barbaresi@unibo.it (A.B.); daniele.torreggiani@unibo.it (D.T.); patrizia.tassinari@unibo.it (P.T.)
2   Department of Industrial Engineering, University of Bologna, Via Terracini 34, 40131 Bologna, Italy; beatrice.pulvirenti@unibo.it
*   Correspondence: marco.bovo@unibo.it

**Abstract:** This paper investigates the aerodynamic behavior of Basil (i.e., *Ocimum basilicum*) and Mentuccia (i.e., *Clinopodium nepeta* (L.) Kuntze), emphasizing the impact of plant structure on drag force. In this paper, the drag coefficient is assessed for the two crop species under various configurations, starting from the pressure drop measured through wind tunnel tests. The methodology involves an innovative use of image processing techniques to determine the leaf area density (LAD) for both Basil and Mentuccia. This approach allows for a precise differentiation between leaf areas and crop pores, crucial for accurate aerodynamic analysis. For Basil, LAD values ranged from 2.41 to 5.08 $m^2 \cdot m^{-3}$, while Mentuccia displayed LAD values between 1.17 and 1.93 $m^2 \cdot m^{-3}$, depending on the crop configuration. This study provides the relationship between plant morphology, canopy density, and drag coefficient, highlighting how these aspects are influenced by different wind velocities. These results are fundamental and necessary for the proper definition of crop behavior and the aerodynamic parameters in Computational Fluid Dynamics simulations. This knowledge is not only fundamental to the field of agricultural aerodynamics but also has significant implications for optimizing crop planting and arrangement, leading to more efficient farming practices and better understanding of plant–environment interactions.

**Keywords:** wind tunnel; LAD; drag coefficient; canopy morphology

## 1. Introduction

The aerodynamic parameters of the crops characterize the effect of the canopy on air flow; these parameters include the drag coefficient and the leaf area density (LAD) [1]. In order to evaluate the impact of crops on the air flow and exchange processes of momentum and climate parameters, including heat, water vapor, and $CO_2$, it is crucial to understand the drag force produced by plants.

Several studies indicate that drag coefficient depends on the type of plant, canopy density, porosity, and shape of leaves and their flexibility. This diversity suggests that a one-size-fits-all approach is insufficient for accurately capturing the complex interactions between plant canopies and air flow. Therefore, the experimental tests available in the literature aiming to determine the drag coefficient have been accomplished through a direct method to calculate the drag force on crops [2] or, in those cases in which it is possible to neglect the impact of crops on turbulent flow, by measuring the velocity and the pressure drop [3].

In greenhouse CFD modeling [4,5], the crop canopy is modeled as a simplified equivalent porous medium, and the canopy effects on the airflow are modeled as an average for the volume of the canopy. The pressure and viscous forces generated by canopy elements, i.e., leaves and branches, result in a rise in the momentum sink, causing a reduction in the

air flow [6]. The drag coefficient, which is necessary to model the crop canopy as an equivalent porous medium, is often considered as a constant of the aerodynamic properties of the crops, irrespective of wind speed or crop species [7]. However, recent studies have begun to challenge this assumption, emphasizing the need for more detailed information regarding how different crop species and varying wind conditions affect the drag coefficient.

On the other hand, it has been observed that the CFD model's reliability is strongly related to the adopted drag coefficient value [8], and the accuracy in the estimation of fluid flow and of downstream air velocity values depend on the assumed drag coefficient. Several studies have investigated the crop influence on air flow inside the greenhouse through wind tunnel measurements. A wind tunnel test is an effective method to investigate the aerodynamics of air flow through vegetation [9,10]. In fact, wind tunnels create a controlled environment that makes it possible to conduct in-depth research on how crop characteristics like porosity and flexibility affect the drag coefficient value [11]. These studies are crucial in filling the gap between theoretical models and practical, real-world applications.

Manickathan's research extended the application of wind tunnel experiments beyond small crops, focusing on trees in urban environments, and investigated how well small-scale tree models can replicate the aerodynamic behavior of larger, real-life trees by comparing their drag coefficients and turbulent flow patterns [12]. Implementing larger vegetation types like trees highlights the wide-ranging applicability and potential of wind tunnel experiments in various environmental research areas.

These highly controlled tests provide us a firm understanding of how crop canopy affects the drag coefficient. Therefore, wind tunnel tests can provide the necessary information to strengthen the reliability of equivalent CFD models, the latter allowing for a more precise evaluation of the effects of vegetation on the microclimate [2,3,10,13]. By enhancing our knowledge of these interactions, such models can become powerful tools in environmental management and urban planning.

Different studies have indicated that the drag coefficient changes with the crop species, the amount of vegetation, and the porosity of the plants [14,15]. Another considered aspect is the impact of changes in the canopy morphology on the airflow field of the spray [16].

Understanding the flow dynamics around and through crops presents a complex challenge, primarily due to the diverse scales of stems and leaves in vegetation [17]. Factors such as variations in plant morphology further add layers of complexity to this issue [18]. These dynamic morphological aspects significantly complicate the quantification of flow resistance presented by vegetation [1,19]. Research by [19] underlines this complexity, demonstrating that the total drag on a single plant is intrinsically linked to its morphological attributes, specifically its leaf area and the ratio of leaf area to stem area.

Thereby, in the present paper, the drag coefficient has been experimentally obtained through tests on two crop species with different morphological features since porosity plays the major role in the aerodynamic effects of the crops. The crop species under study are two small aromatic plants, Basil (i.e., *Ocimum basilicum*) and Mentuccia (*Clinopodium nepeta* (L.) Kuntze). This choice of species allows for an investigation into how different plant structures, particularly in terms of leaf configuration and canopy density, impact aerodynamic parameters. This research aims to provide more insight into the accurate aerodynamic characterization of vegetation useful for the calibration of equivalent volume models to be used for solving complex CFD simulations. This is especially relevant in scenarios where precision is crucial, such as in the analysis of wind flow in urban green spaces or in smart farming. The approach adopted in this paper is significantly general, and the results can be highly effective in defining a pipeline for CFD simulations related to crop modeling. It is particularly applicable in scenarios such as plant production in greenhouse buildings, analyzing the behavior of green infrastructures, and designing nature-based solutions. These applications demonstrate the broad potential impact of this research in both agricultural and urban environmental contexts.

## 2. Materials and Methods

### 2.1. Theoretical Considerations

Crops could be considered a porous medium, characterized by a solid matrix with interconnected pores. The flow within the porous medium is described by Darcy's law, which relates the pressure gradient $\nabla p$ to the volume-averaged velocity vector ($u$) in the control volume.

$$\nabla p = -\phi \frac{\mu}{K} u \tag{1}$$

where $K$ is the permeability of the porous medium, which is independent of the nature of the fluid and depends on the pore geometry in the medium; $\phi$ is the porosity of the medium; and $\mu$ is the viscosity of the air. This relationship forms the basis of modeling the movement of air through crop canopies, allowing for the prediction of airflow behavior in agricultural environments. While Darcy's law represents Newton's second law of motion, its limitation lies in the absence of the inertial term $u^2$. This limitation becomes particularly noticeable at higher fluid velocities, where inertial effects cannot be ignored. To address this issue, the Darcy–Forchheimer equation is used, incorporating a quadratic term.

$$\nabla p = \left( \frac{\mu}{K} u + \frac{C_f \, \rho}{K^{1/2}} u^2 \right) \tag{2}$$

The equation introduces a dimensionless inertial factor $C_f$, which represents the nonlinear momentum loss coefficient and the fluid density $\rho$ (kg m$^{-3}$). This enhancement allows for a more comprehensive modeling of airflow, especially in scenarios involving turbulent conditions. Notably, the value of $C_f$ depends on the airflow direction with respect to the leaves. To determine the validity domains of the Darcy and Darcy–Forchheimer laws, a modified Reynolds number $Re_m$ is employed.

$$\mathrm{Re}_m = \frac{u \, K^{1/2}}{v} \tag{3}$$

where $v$ is the kinematic viscosity of the fluid (m$^2$ s$^{-1}$), $u$ is the fluid velocity (m s$^{-1}$), and $K^{1/2}$ corresponds to the characteristic dimension of the porous medium (i.e., dimension of the average pore). The application of this Reynolds number helps in identifying the appropriate model to use based on the specific conditions of airflow through the crop canopy. The Darcy law is valid when $Re_m < 1$. However, if $Re_m > 1$, the linear term is not constant and can vary based on the medium's porosity and the characteristic air speed. This variation is crucial for accurately predicting airflow in more turbulent conditions typically found in outdoor agricultural environments. For high Reynolds numbers $Re_m > 10$, the momentum sink can be expressed as a source term in the Navier–Stokes Equation [20].

$$\nabla p = -\rho \, LAD \, C_D \, u^2 \tag{4}$$

where $LAD$ is the leaf area density (m$^2$ m$^{-3}$) and represents the ratio of the total leaf area to the total volume of the crop cover, $C_D$ is the drag coefficient and can be defined as the ratio of the pressure difference of windward and leeward and the dynamic force [21], and u is the air velocity. Integrating LAD into the Navier–Stokes equation allows for a more precise estimation of the impact of leaf density on airflow dynamics. By combining the equations related to the Darcy–Forchheimer Equation (2) and the drag effect Equation (4), it provides:

$$\frac{C_f}{K^{1/2}} = LAD \, C_D \tag{5}$$

Then, the calculation of the drag coefficient $C_D$ can be achieved through the measurement of pressure loss across a vegetation canopy for various air velocities. This methodology is key for accurately quantifying the aerodynamic resistance offered by different crop types, which is essential for precise simulations and predictions in CFD models. These

models can then be applied to a variety of practical scenarios, ranging from optimizing crop arrangements for airflow in greenhouses to improving the design of urban green spaces for enhanced microclimate control.

### 2.2. Wind Tunnel Tests

An open circuit wind tunnel built in the Department of Industrial Engineering of the University of Bologna has been used for carrying out the experimental tests. The utilization of a wind tunnel enables the possibility to have a regulated environment in terms of temperature and airflow velocity. This control is crucial to ensure that the tests reflect a consistent and repeatable set of environmental conditions, eliminating external variables that could affect the results. As shown in Figure 1, the wind tunnel test chamber has dimensions of $30 \times 30 \times 60$ cm . The wind tunnel parts start with a contraction section of a 90 cm honeycomb as an inlet, attached to a channel that leads directly to the test chamber. This design facilitates a smooth and uniform airflow into the test section, which is essential for accurate aerodynamic measurements. A narrow channel follows the measurement chamber connected to the fan, the outlet of the wind tunnel. The airflow in this system is provided by the fan having a diameter of 45 cm with output rotation frequency ranging from 0 Hz to 50 Hz. The measurements in the wind tunnel were performed in order to calculate the drag coefficient $C_D$ of the crops.

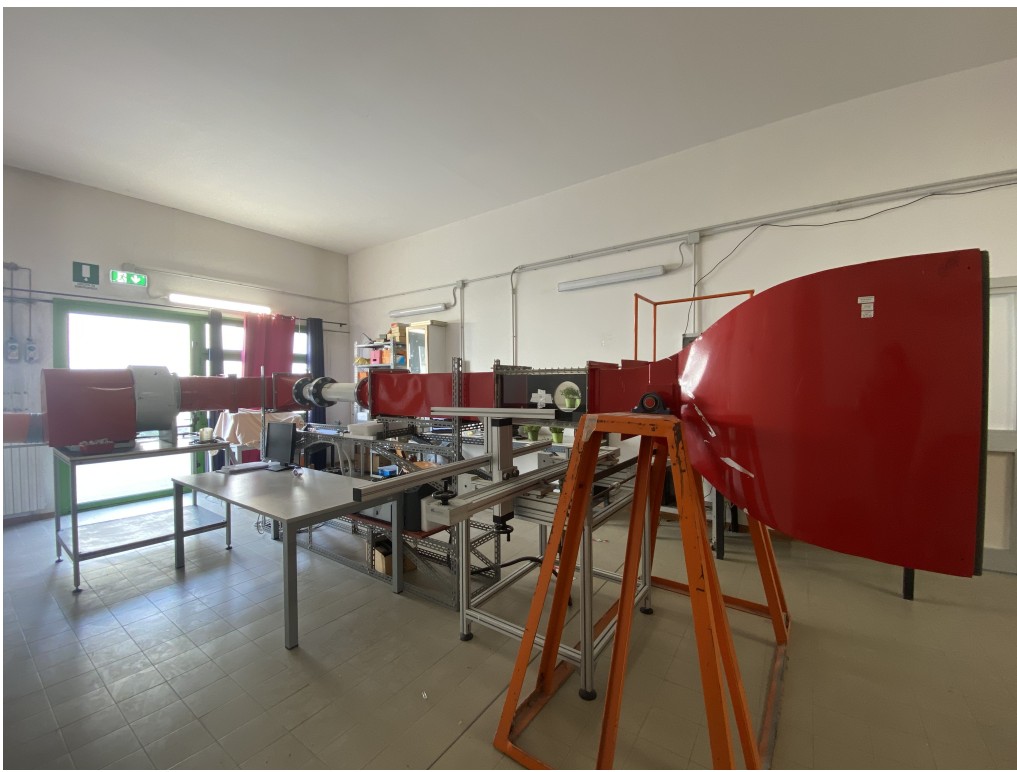

**Figure 1.** Wind tunnel used for the tests on the crops.

A calibration curve of the system was determined before starting the measurements on crops in order to understand the relation between the fluid (i.e., air) velocity and the fan rotation frequency and also to ensure the accuracy of the velocity measurements and to establish a reliable baseline for interpreting the experimental data.

The measurements were started after reaching a constant airflow. Twenty-four readings of static pressure and dynamic pressure have been collected using a Pitot tube probe and a micromanometer (DP-Calc Micromanometer, Model 8715) with an accuracy of $\pm 0.01$ ms$^{-1}$. The measurement interval was 5 s. This high level of precision in pressure measurements is essential for calculating the drag coefficient with the necessary accuracy. The measurements were carried out under different frequencies of the fan, beginning from

5 Hz and increasing by 5 Hz every reading until it reached 30 Hz. This range of frequencies was chosen to simulate a variety of wind conditions that crops might encounter in horticultural settings. For each test point, the mean value of the 24 collected datums has been calculated and then considered. Thirty readings of air velocity measurements at a 2 s interval were obtained by the same instrument and were measured by placing the Pitot tube upstream, 190 mm away from the test chamber, and also 190 mm downstream. This positioning ensures a comprehensive assessment of the airflow both before it encounters the crop canopy and after it has passed through, offering insights into the aerodynamic effects of the crops.

The experimental investigation involved three phases aimed at studying the density variation and analyzing the irregular shape and porosity of the canopy. Each phase involved different configurations of crops within the test section. In the first phase, a single Basil pot was placed in the test section, and measurements were performed at the leaves' level (see Figure 2). This initial phase provides baseline data on how a single plant affects airflow, which is fundamental for understanding the cumulative effects observed in later phases. To assess the impact of different frontal areas, the Basil pot was rotated by 90°, and the measurements were repeated. The second phase consisted of two Basil pots placed in the test section, while the third phase involved three pots within the test chamber. Similar to the first phase, measurements were taken at the same point, and crop rotation was implemented to treat the combined crops as a canopy. The subsequent phases allowed for the observation of how increasing canopy density influences aerodynamic properties.

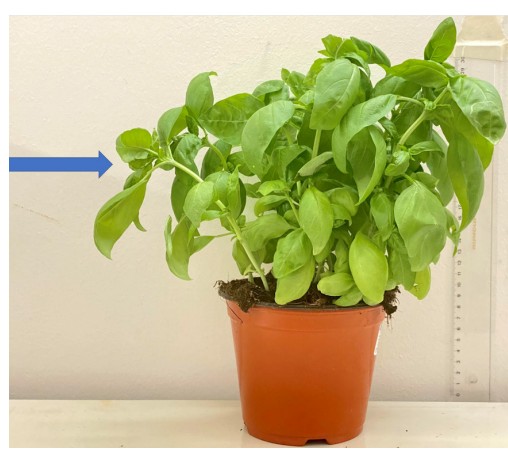 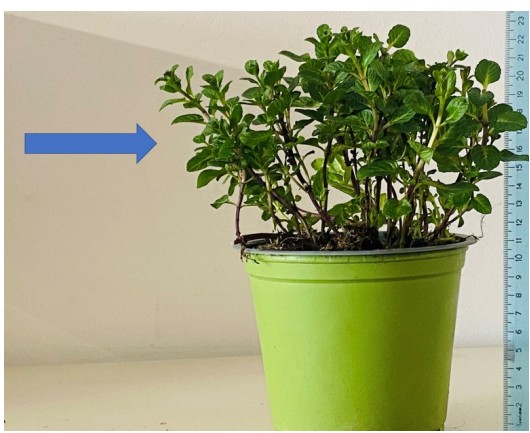

(**a**) Example of Basil crop.          (**b**) Example of Mentuccia crop.

**Figure 2.** Measurement positions indicated by blue arrays for the two crops investigated.

To ensure optimal conditions for the crops, they were watered up to field capacity until the day prior to the test. The measurement setup was conducted rapidly to minimize the possibility of wilting leaves and to maintain the health of the crops throughout the experiments. This consideration is critical, as the physical state of the plants could significantly impact the aerodynamic properties being measured. The entire experimental setup was then replicated for the Mentuccia crop to enable a comparative analysis. This replication with a different species provides a broader understanding of how various types of vegetation interact with airflow, enhancing the applicability of the study's findings.

### 2.3. Image Elaboration

The porosity value has been determined by analyzing the image through the use of ImageJ v1.46 software [22], which has proven effective in measuring leaf area [23]. ImageJ is known for its versatility and precision in image analysis and offers a range of tools that are ideal for quantifying complex plant structures. Prior to conducting the wind tunnel tests, images were captured to distinguish the frontal area of the crops in a still-air environment.

This step is crucial as it ensures that the plant's natural posture and structure are accurately represented in the analysis.

These images were subsequently processed and a calibration procedure was performed to establish the image size, followed by setting a color threshold to differentiate the crop from the background. This is essential for maintaining consistency across all images, thus allowing for a reliable comparative analysis. Non-target noise pixels were removed to enhance the accuracy of the analysis, as these extraneous pixels can significantly skew the quantification of porosity. Meanwhile, the crop itself was extracted. Figure 3 shows the image processing algorithm in detail. The resulting image was then converted into a binary image, separating it into two distinct regions: a black area representing the crops and a white area representing the pores within the crops (see Figures 4 and 5). This binary conversion simplifies the complex structures of the crops into a more analyzable form, making it possible to precisely calculate the porosity. This binary representation facilitated the assessment of porosity by quantifying the ratio of the white area (pores) to the total area of the image. Through this method, this approach provides a clear and quantifiable measure of the porous nature of the crops, which is a key factor in understanding their aerodynamic behavior.

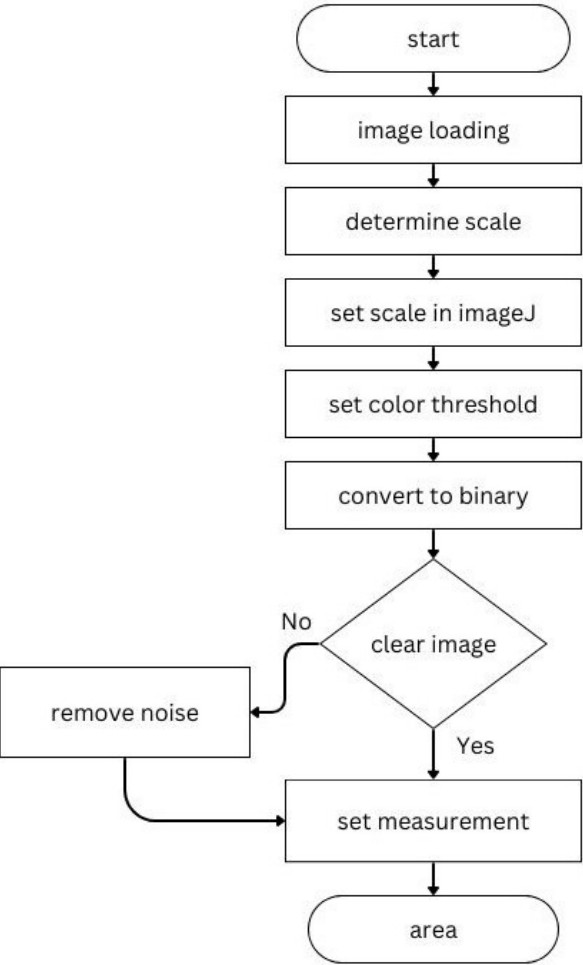

**Figure 3.** General scheme for the image processing.

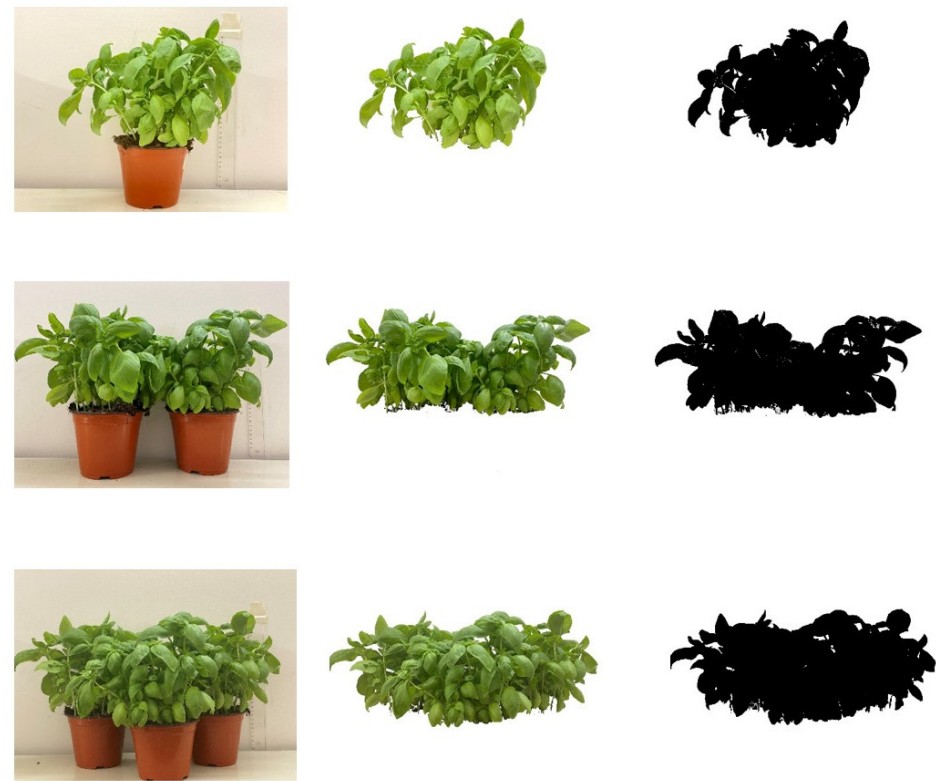

**Figure 4.** Example of image elaboration process for different set of Basil crops.

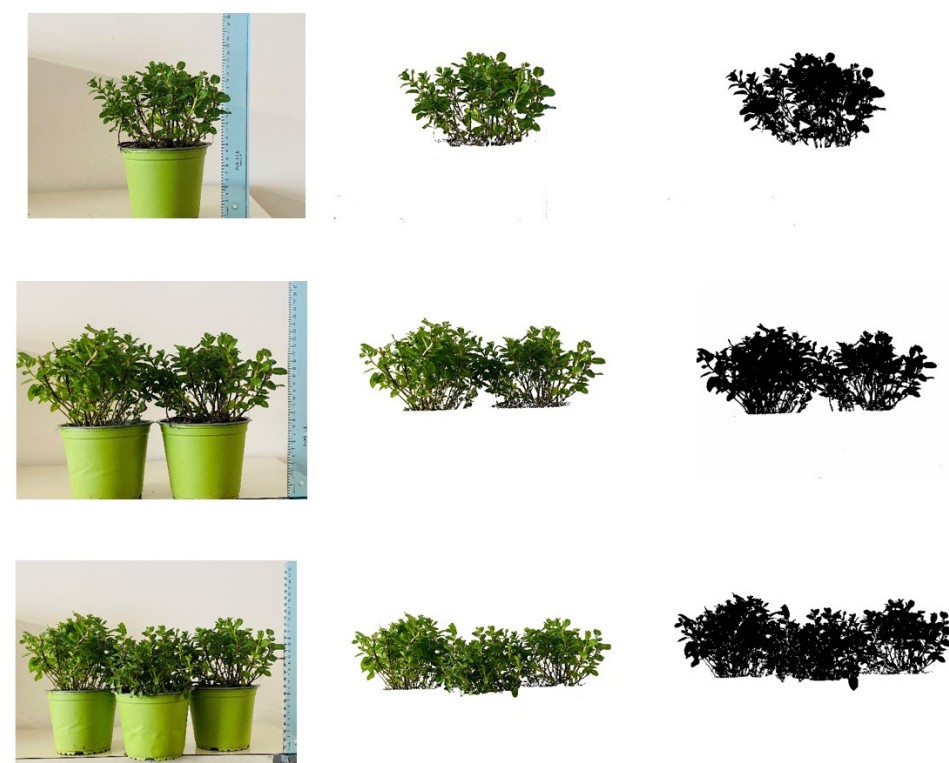

**Figure 5.** Example of image elaboration process for different set of Mentuccia crops.

## 3. Results

### 3.1. Leaf Area Density Assessment

Leaf area density was calculated as the ratio between leaf area per unit of vertical surface area and the sample thickness in the flow direction. This calculation involved a novel image elaboration technique, where the leaf area was determined from the binary image that differentiates between the crop leaf area and the pores within the crop. This technique not only enhances the precision of the measurement but also provides a more detailed understanding of the spatial distribution of the leaf area.

For the Basil crop, the LAD values showed a direct relationship with plant density, indicating significant variations as the number of pots in the test chamber increased. Specifically, the LAD was measured at 2.41 $m^2$ $m^{-3}$ for a singular pot, which then increased to 3.07 $m^2$ $m^{-3}$ with the addition of the second pot, and further increased to 5.08 $m^2$ $m^{-3}$ upon the addition of the third pot. These findings are detailed in Table 1 and suggest a strong relationship between plant density and leaf area distribution. In contrast, the Mentuccia crop showed a clearly different pattern of LAD variation. Starting at a lower LAD of 1.17 $m^2$ $m^{-3}$ for one pot, it increased to 1.64 $m^2$ $m^{-3}$ with two pots and 1.93 $m^2$ $m^{-3}$ with three pots. This moderate increase indicates a less significant influence of plant density on the airflow dynamics and might be attributed to the crop's naturally less dense leaf arrangement.

**Table 1.** LAD values and standard deviation for Basil and Mentuccia. Test 1 represents the case with 1 pot, Test 2 represents the case with 2 pots, and Test 3 represents the case with 3 pots.

| Crop | LAD ($m^2$ $m^{-3}$) | | | Standard Deviation ($m^2$ $m^{-3}$) |
|---|---|---|---|---|
| | Test 1 | Test 2 | Test 3 | |
| Basil | 2.41 | 3.07 | 5.08 | ±1.39 |
| Mentuccia | 1.17 | 1.64 | 1.93 | ±0.38 |

### 3.2. Drag Coefficient

Experimental wind tunnel studies investigating drag coefficients have demonstrated the interaction between plant morphology, canopy density, and aerodynamic resistance across different air velocities.

The Basil crop displayed minimal variability in airflow measurements at lower velocities, which significantly expanded at higher velocities, indicating a complex interaction between plant structural characteristics and aerodynamic forces.

Conversely, the Mentuccia crop demonstrated a consistent pattern of variability across all tested velocities. This consistency across different airflow conditions, as shown in Figure 6, suggests a uniform aerodynamic response that could be valuable for the plant's adaptation to varying environmental conditions.

The drag coefficient calculations for each crop, based on velocity and pressure drop data, are presented in Figure 7. The Basil's drag coefficient showed a substantial increase from 0.03 to 0.45 as air velocity increased, whereas the Mentuccia's drag coefficient ranged more broadly from 0.10 to 0.75. These variations highlight the dynamic interaction between plant structures and aerodynamics.

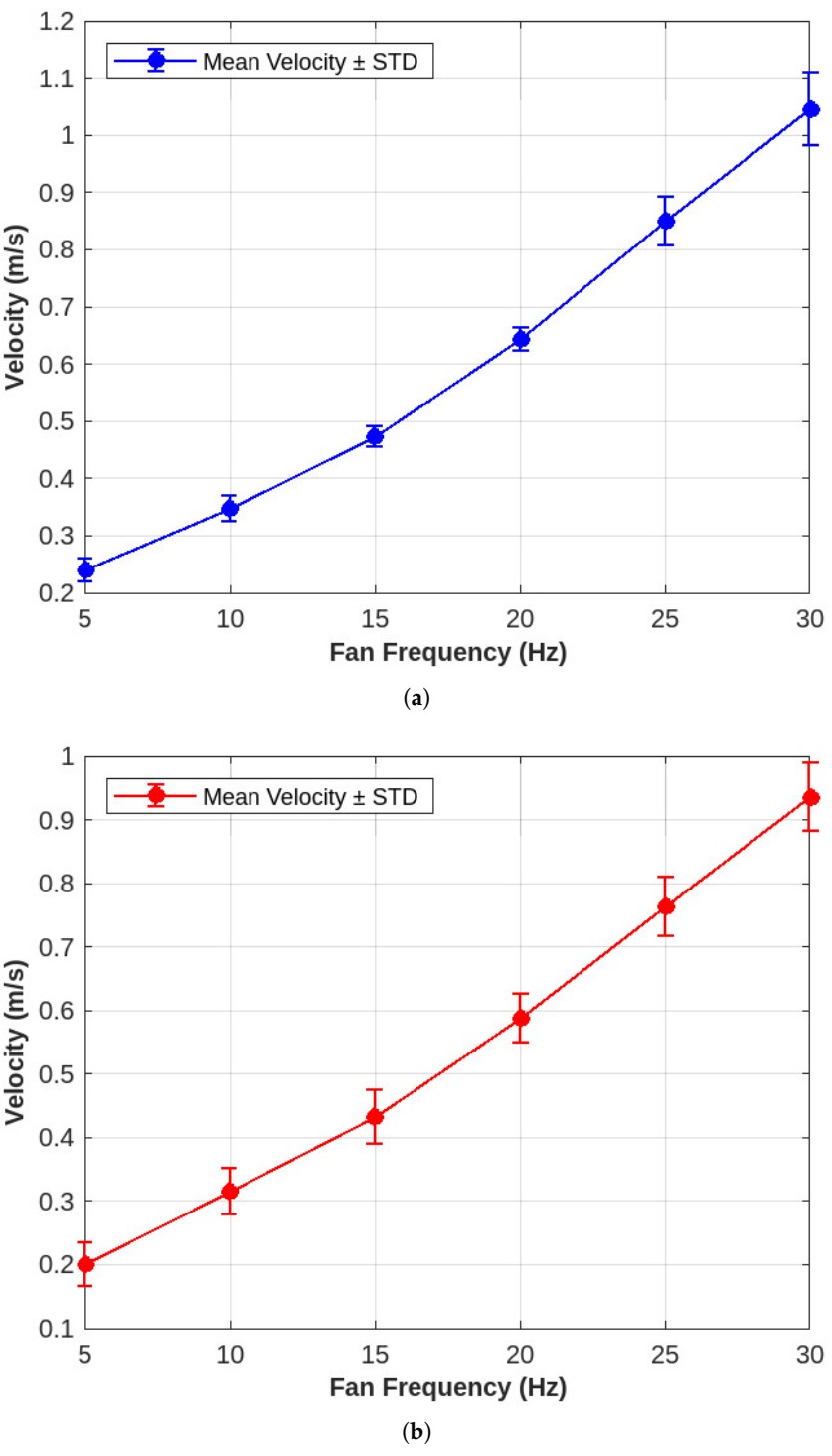

**Figure 6.** Error analysis between air velocity and fan frequency for (**a**) Basil and (**b**) Mentuccia crops. Points represent mean velocities, and error bars show standard deviation.

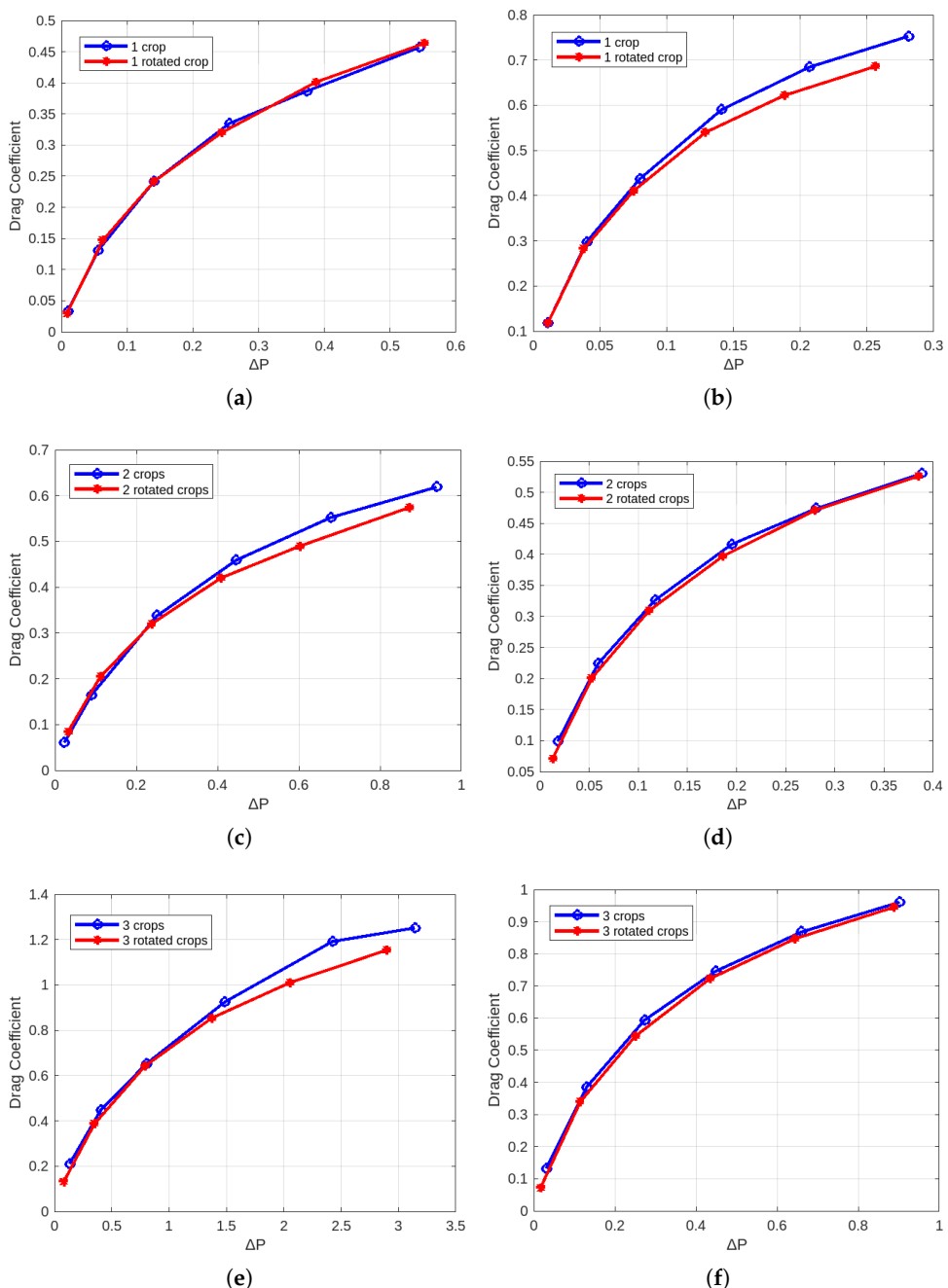

**Figure 7.** Pressure drop (in Pa) versus drag coefficient for the frontal and 90°-rotated side for Basil and Mentuccia crops with different numbers of pots in the test chamber. (**a**) One pot of Basil; (**b**) one pot of Mentuccia; (**c**) two pots of Basil; (**d**) two pots of Mentuccia; (**e**) three pots of Basil; (**f**) three pots of Mentuccia.

## 4. Discussion

The variation in LAD with plant density highlights the sensitivity of airflow dynamics to vegetation density. The Basil crop's increasing LAD with more pots suggests a cumulative effect on aerodynamic parameters, which could significantly influence airflow through the vegetation. This effect is less noticeable in the Mentuccia crop, possibly due to its unique plant morphology and less dense leaf arrangement. A similar result has been reported by [24], which indicated that plant morphology affects airflow and can reduce wind erosion.

Peruzzo et al. [25] highlighted how airflow dynamics are affected by the properties of vegetation, showing that drag coefficients are significantly influenced by factors such

as the submergence ratio and stem density, with these effects being more dependent on the vegetation's physical structure than on the airflow conditions. This insight supports our findings regarding the correlation between canopy density and drag coefficients in both plants.

The relationship between canopy density and drag coefficient was clear in both crops, with denser canopies resulting in higher drag coefficients, aligning with the findings from [26]. This correlation indicates a significant impact of canopy structure on airflow resistance. Moreover, the plants' flexibility influences their aerodynamic behavior. Basil's flexibility allowed it to adapt to airflow, reducing drag, whereas Mentuccia's rigidity presented a greater obstacle to airflow, as supported by the research of [27].

The observed variation in drag coefficient with crop rotation and air velocity challenges traditional assumptions of a uniform drag coefficient across different plant types and conditions. This variation emphasizes the need for careful selection and calibration of drag coefficients in (CFD) analyses to achieve accurate simulations, especially in complex environmental interactions that involve vegetation, as previously suggested by [3,28].

The findings from this study contribute to a deeper understanding of plant and aerodynamic interactions, offering valuable insights for agricultural and urban planning applications sensitive to wind effects. These insights highlight the importance of considering the specific characteristics of each species and plant orientation in modeling to accurately predict airflow patterns in vegetated environments.

Further research could study the quantification of LAD and drag coefficient across different plant species and environmental conditions. Such studies would provide a more comprehensive understanding of the factors influencing plant–aerodynamic interactions, supporting the development of more accurate environmental and agricultural models.

## 5. Conclusions

This study has provided valuable insights into the aerodynamic behaviors of Basil and Mentuccia crops, particularly in relation to leaf area density (LAD) and its impact on drag coefficient. The results indicate that variations in LAD are crucial in understanding how different plant structures interact with airflow. Basil, with its higher LAD, demonstrates adaptability to airflow changes due to its flexible canopy, resulting in a lower drag coefficient. Mentuccia, with a more gradual LAD increase and firmer structure, shows less variation in drag coefficient, underscoring the importance of plant morphology beyond leaf density and airflow conditions in determining aerodynamic behavior.

These insights are important for agricultural optimization, suggesting that crop arrangement and selection based on aerodynamic principles can enhance photosynthetic efficiency and reduce wind stress. However, this study faced limitations due to the experimental conditions and the size of the test chamber affecting the generality of results. This research emphasizes the potential of applying aerodynamic data in precision agriculture.

**Author Contributions:** Conceptualization, S.H.F.A.-R., M.B., E.S., B.P. and D.T.; methodology, S.H.F.A.-R., M.B., E.S. and B.P.; data curation, S.H.F.A.-R., M.B. and E.S.; writing—original draft preparation, S.H.F.A.-R., M.B. and B.P.; writing—review and editing, S.H.F.A.-R., M.B., B.P. and A.B.; supervision, M.B., B.P., A.B., D.T. and P.T.; project administration, D.T. and P.T. All authors have read and agreed to the published version of the manuscript.

**Funding:** This research received no external funding.

**Institutional Review Board Statement:** Not applicable.

**Data Availability Statement:** The data presented in this study are available on request from the corresponding author.

**Conflicts of Interest:** The authors declare no conflicts of interest.

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
