# Peer review of "Assessment of the Influence of Canopy Morphology on Leaf Area Density and Drag Coefficient by Means of Wind Tunnel Tests"

_sustainability, doi:10.3390/su16052010_

Round 1
Reviewer 1 Report
Comments and Suggestions for Authors
1. The use of wind tunnel tests and image processing for quantifying LAD and drag coefficients is comprehensive. However, the size of the wind tunnel and the limitations in crop species used could affect the generalizability of the results. I would suggest adding more justification regarding the selection of the wind tunnel size and crop species.
2. I would recommend authors add a standard deviation for LAD.
3. Some words are too vague, for example, “To ensure optimal conditions for the crops, they were proper watered until the day prior to the tests.” It would be better to add a rough amount instead of using proper to describe.
Overall, this article has a clear structure and objectives, and the pointed-out limitations provide avenues for future research.
Author Response
The Authors are grateful to the Reviewers for their valuable comments, because thank to these, some parts in the paper have been improved and added, together with the addition of some references showing that the theme and the methodology presented in this paper is currently of interest for academic research. For this reason, we hope that the revised version of the paper could be suitable for publication in Sustainability.
We reported below the reviewer comments and our replies (in Italic).
- The use of wind tunnel tests and image processing for quantifying LAD and drag coefficients is comprehensive. However, the size of the wind tunnel and the limitations in crop species used could affect the generalizability of the results. I would suggest adding more justification regarding the selection of the wind tunnel size and crop species.
1) The authors thank the Reviewer for the comment. The authors would like to underline that this is a first study on the subject. The investigations have been limited to two crop species characterized by different leaves sizes, density, numerosity etc, but considered representative of several species in the aromatic family. On the other hand, the authors have used the available wind tunnel and this equipment typology is commonly used for this type of investigations since it ensures velocity range and Reynolds number, suitable for these investigations. The wind tunnel equipment has been calibrated before the tests and the authors believe that the dimensions do not constitute a limiting factor for this type of research since several other aromatic crops have similar dimensions and morphological properties. Anyway, we have added in the manuscript some remarks about the abovementioned aspects.
- I would recommend authors add a standard deviation for LAD.
2) The standard deviation has been added as requested by the Reviewer.
- Some words are too vague, for example, “To ensure optimal conditions for the crops, they were proper watered until the day prior to the tests.” It would be better to add a rough amount instead of using proper to describe.
3) The sentence has been carefully revised. Moreover, the paper has been checked and revised following the indication of the Reviewer and the structure of the Journal.
Overall, this article has a clear structure and objectives, and the pointed-out limitations provide avenues for future research.
- R) The Authors thank the Reviewer for the comment.
Reviewer 2 Report
Comments and Suggestions for Authors
The article "Assessment of the Influence of Canopy Morphology on Leaf Area Density and Drag Coefficient by means of Wind Tunnel Tests " presents very important aspects, especially for agriculture and botany. The authors introduce the topic very well. Charts and photos diversify the publication. The authors also appropriately present their results and their meaning in practice. The only comment I have is that I didn't notice in the discussion part, references to other research and literature, which could enrich the work. In my opinion, the publication could be published after this minor correction.
1. What is the aerodynamic behavior of basil and Mentuccia, highlighting the influence of the plant structure on the drag force.
2. The most important part is the research part carried out using a wind tunnel, which provides information that can be used in practice.
3. These studies are important both in the agricultural aspect (adjustment of cultivation conditions) and urban landscape development, which will allow for large savings in the future in terms of selecting good vegetation or conditions right away, rather than searching by trial and error.
4. I have no objections to the methodology, but to the discussion and bibliography.
5. I do not have any comments on the conclusions, but on emphasizing them by comparing my study with other studies.
6. The bibliography is adequate but could be more extensive to add validity and scientific credibility to the publication
7. The figures are correct, no comments.
Author Response
The Authors are grateful to the Reviewers for their valuable comments, because thank to these, some parts in the paper have been improved and added, together with the addition of some references showing that the theme and the methodology presented in this paper is currently of interest for academic research. For this reason, we hope that the revised version of the paper could be suitable for publication in Sustainability.
We reported below the reviewer comments and our replies (in Italic).
--------------------------------------------------------------------------
The article "Assessment of the Influence of Canopy Morphology on Leaf Area Density and Drag Coefficient by means of Wind Tunnel Tests " presents very important aspects, especially for agriculture and botany. The authors introduce the topic very well. Charts and photos diversify the publication. The authors also appropriately present their results and their meaning in practice. The only comment I have is that I didn't notice in the discussion part, references to other research and literature, which could enrich the work. In my opinion, the publication could be published after this minor correction.
- R) The Authors thank the Reviewer for the comment. Following the suggestion of the reviewer in the revised version of the paper a comparison with results available in literature has been added.
- What is the aerodynamic behavior of basil and Mentuccia, highlighting the influence of the plant structure on the drag force.
- The most important part is the research part carried out using a wind tunnel, which provides information that can be used in practice.
- These studies are important both in the agricultural aspect (adjustment of cultivation conditions) and urban landscape development, which will allow for large savings in the future in terms of selecting good vegetation or conditions right away, rather than searching by trial and error.
- I have no objections to the methodology, but to the discussion and bibliography.
- I do not have any comments on the conclusions, but on emphasizing them by comparing my study with other studies.
The Authors thank the Reviewer for the frank opinion and suggestions. The paper has been checked and revised following the indication of the Reviewer and the structure of the Journal. Some references have been added in the results section and the discussion section has been improved.
Reviewer 3 Report
Comments and Suggestions for Authors
This paper explores the aerodynamic behavior of Basil (Ocimum basilicum) and Mentuccia (Clinopodium nepeta (L.) Kuntze), with a specific focus on the impact of plant structure on drag force. Investigating the aerodynamics of crop species is relevant and can contribute to our understanding of agricultural processes.
The paper effectively communicates its objectives, methods, and findings. However, providing more details about the wind tunnel test setup and the image processing techniques used for Leaf Area Density (LAD) determination will be good.
Including specific information about the wind tunnel parameters (e.g., airflow rates, dimensions) and a detailed description of the image processing algorithms is necessary
To improve transparency, consider including error analysis or uncertainties associated with the measurements, for example in figure 5.
While the study mentions the relationship between plant morphology, canopy density, and drag coefficient, it would be valuable to delve deeper into the interpretation of these relationships.
Additionally, discussing how various wind velocities affect the findings would add depth to the discussion.
The paper should address the practical implications of the findings for agriculture. For instance, how can farmers use this information to optimize crop configurations and improve aerodynamic performance?
Offering suggestions for future research or practical applications would round out the conclusion section effectively.
There are a few grammatical and formatting issues that require attention. A final proofreading pass is recommended to ensure clarity and consistency.
Comments on the Quality of English LanguageThere are a few grammatical and formatting issues that require attention. A final proofreading pass is recommended to ensure clarity and consistency.
Author Response
The Authors are grateful to the Reviewers for their valuable comments, because thank to these, some parts in the paper have been improved and added, together with the addition of some references showing that the theme and the methodology presented in this paper is currently of interest for academic research. For this reason, we hope that the revised version of the paper could be suitable for publication in Sustainability.
We reported below the reviewer comments and our replies (in Italic).
--------------------------------------------------------------------------
This paper explores the aerodynamic behavior of Basil (Ocimum basilicum) and Mentuccia (Clinopodium nepeta (L.) Kuntze), with a specific focus on the impact of plant structure on drag force. Investigating the aerodynamics of crop species is relevant and can contribute to our understanding of agricultural processes.
R) The Authors thank the Reviewer for the comment.
The paper effectively communicates its objectives, methods, and findings. However, providing more details about the wind tunnel test setup and the image processing techniques used for Leaf Area Density (LAD) determination will be good.
R) The authors have added some further details as requested by the Reviewer.
Including specific information about the wind tunnel parameters (e.g., airflow rates, dimensions) and a detailed description of the image processing algorithms is necessary
R) Following the indication of the Reviewer, the authors have explained in detail the parameters of the wind tunnel including the airflow rates and the main dimensions. Regarding the image processing techniques, the authors have added a general scheme describing the process algorithm as suggested by the Reviewer.
To improve transparency, consider including error analysis or uncertainties associated with the measurements, for example in figure 5.
R) The authors have included the error analysis of air velocity measurements and fan frequency as suggested by the Reviewer.
While the study mentions the relationship between plant morphology, canopy density, and drag coefficient, it would be valuable to delve deeper into the interpretation of these relationships.
R) The authors have explained with more detail the relationship between crop morphology, canopy density, and drag coefficient.
Additionally, discussing how various wind velocities affect the findings would add depth to the discussion.
R) The effect of different wind velocities has been included in the paper as suggested by the Reviewer.
The paper should address the practical implications of the findings for agriculture. For instance, how can farmers use this information to optimize crop configurations and improve aerodynamic performance?
R) The practical implications have been included in the paper as suggested by the reviewer.
Offering suggestions for future research or practical applications would round out the conclusion section effectively.
R) The authors have also added some suggestions for future research and practical applications in the conclusion section.
There are a few grammatical and formatting issues that require attention. A final proofreading pass is recommended to ensure clarity and consistency.
R) The Authors thank the Reviewer for the comment. The text has been checked as a whole following the indication of the Reviewer.
Round 2
Reviewer 3 Report
Comments and Suggestions for Authors
The authors have made significant improvements to the manuscript. However, for publication in Sustainability, substantial revisions are still required based on the following points:
The Introduction relies on outdated citations. It should be updated with more recent references, preferably within the last 5-10 years.
The Discussion section is inadequately supported, featuring only 3-4 references. It is recommended to separate the Results and Discussion sections and include additional material with appropriate citations.
The Conclusion section is overly lengthy. Authors should aim to succinctly conclude their study within one paragraph, spanning approximately 100-150 words.
There are errors in the English language, particularly in the use of punctuation. For example, I have corrected mistakes in the first paragraph of the introduction below. The entire manuscript should undergo revision with the assistance of a native English speaker.
“The aerodynamic parameters of the crops characterize the effect of the canopy on air flow; these parameters include the drag coefficient and the leaf area density (LAD) [1]. In order to evaluate the impact of crops on the airflow and exchange processes of momentum and climate parameters including, heat, water vapor, and CO2, it is crucial to understand the drag force produced by plants.”
Comments on the Quality of English LanguageThere are errors in the English language, particularly in the use of punctuation. For example, I have corrected mistakes in the first paragraph of the introduction below. The entire manuscript should undergo revision with the assistance of a native English speaker.
“The aerodynamic parameters of the crops characterize the effect of the canopy on air flow; these parameters include the drag coefficient and the leaf area density (LAD) [1]. In order to evaluate the impact of crops on the airflow and exchange processes of momentum and climate parameters including, heat, water vapor, and CO2, it is crucial to understand the drag force produced by plants.”
Author Response
Dear Editor,
We extend our sincere gratitude to you for managing the submission process and to the anonymous reviewers for their insightful feedback. Following their suggestions, we have updated our manuscript, significantly enhancing its quality compared to the initial submission.
We are particularly thankful for the reviewers' constructive comments, which have led to significant improvements in various sections of the paper, including the integration of new references. These updates underscore the relevance of our paper's theme and methodology to ongoing academic research. Consequently, we hope that this revised version aligns well with the Journal of Sustainability publication criteria.
We reported below the reviewer comments and our replies. We attached also a version of the revised manuscript with tracked changes. The modifications have been highlighted.
--------------------------------------------------------------------------------------
Reviewer #3
The authors have made significant improvements to the manuscript. However, for publication in Sustainability, substantial revisions are still required based on the following points:
R) The Authors thank the Reviewer for the comment. The paper has been checked and revised following the indication of the Reviewer and the structure of the Journal. A detailed response to the comments is reported below.
The Introduction relies on outdated citations. It should be updated with more recent references, preferably within the last 5-10 years.
R) More recent references have been added in the Introduction section as suggested by the Reviewer.
The Discussion section is inadequately supported, featuring only 3-4 references. It is recommended to separate the Results and Discussion sections and include additional material with appropriate citations.
R) The Results and Discussion section has been separated, and results supported references have been added in the revised Result section as suggested by the Reviewer.
The Conclusion section is overly lengthy. Authors should aim to succinctly conclude their study within one paragraph, spanning approximately 100-150 words.
R) The Conclusion section has been summarized as suggested by the Reviewer.
There are errors in the English language, particularly in the use of punctuation. For example, I have corrected mistakes in the first paragraph of the introduction below. The entire manuscript should undergo revision with the assistance of a native English speaker.
R) The authors have undertaken a comprehensive proofreading and correction process as suggested by the Reviewer.
Round 3
Reviewer 3 Report
Comments and Suggestions for Authors
Authors are advised to adhere to the formatting conventions of previously published articles in this journal or other academic publications. The Results section should exclusively present the outcomes derived from the current investigation, without including any citations. Conversely, the Discussion section is expected to interpret these findings through the lens of existing literature, incorporating appropriate and pertinent references. It is unusual for the current manuscript to include a few references within the Results section while entirely omitting citations from the Discussion section. Authors must justify and elucidate the implications of their findings by grounding their discussion in relevant and authoritative sources. The discussion section is too short.
Comments on the Quality of English LanguageMinor editing of English language required
Author Response
|